# CutSharp: A Simple Data Augmentation Method for Learned Image Compression

## Abstract

Learned image compression (LIC) methods have demonstrated superior rate–distortion performance, compared to traditional ones. Previous studies on LIC have mainly focused on models, consisting of analysis/synthesis transformations and entropy models. Unfortunately, the importance of *data* has usually been neglected when training LIC models. In this paper, we introduce block-wise RGB standard deviation as a measure for estimating the compression-related difficulty of images. Next, we emphasize the significance of effective data utilization for LIC by demonstrating that models trained on a certain subset of data, constructed according to the block-wise RGB standard deviation, can achieve superior rate–distortion performance to models trained on the entire data. Inspired by this observation, we propose a simple data augmentation technique for LIC, coined `CutSharp`, which enhances image sharpness within an arbitrary region. Our proposed augmentation consistently improves rate–distortion performance on the Kodak and CLIC validation dataset. We hope that our work will encourage further research in data-centric approaches for LIC.

## 1 Introduction

Learned image compression (LIC) (Ballé et al., 2017) has emerged as a prominent area of research, surpassing traditional rule-based approach such as JPEG (Wallace, 1992) and BPG (Bellard, 2015). In general, LIC involves a transform coding framework using learned nonlinear transformations and an learned entropy model (Ballé et al., 2016b). For the encoding process, an image $x$ is transformed into a latent representation $y$ through a nonlinear *analysis transformation* $f_a(\cdot)$. This latent representation $y$ is then quantized into $\hat{y}$, represented as $\hat{y} = \lfloor y \rceil$, and subsequently encoded into a bitstream using an entropy coding algorithm (e.g., arithmetic coding (Rissanen & Langdon, 1981)). To perform entropy coding, an entropy model which serves as a prior probability model on $\hat{y}$ is required. For the decoding process, the bitstream is decoded into the quantized latent representation $\hat{y}$. Finally, a reconstructed image $\hat{x}$ is produced from $\hat{y}$ through a nonlinear *synthesis transformation* $f_s(\cdot)$.

Within this framework, the rate–distortion optimization is formulated as the minimization of both the expected length of the bitstream (i.e., rate) and the expected distance, denoted as $d$, between the original image $x$ and its reconstructed image $\hat{x}$ (i.e., distortion). This can be expressed as follows:

$$\underbrace{\mathbb{E}_{x \sim p_x}\left[-\log_2 p_{\hat{y}}(\lfloor f_a(x) \rceil)\right]}_{\text{rate}} + \lambda \cdot \underbrace{\mathbb{E}_{x \sim p_x}\left[d(x, f_s(\lfloor f_a(x) \rceil))\right]}_{\text{distortion}}. \tag{1}$$

Numerous previous studies have concentrated on designing learned nonlinear transformations and learned entropy models to enhance rate–distortion performance. For better nonlinear transformations, various architectures (e.g., CNN and Transformer) (Ballé et al., 2017; Zou et al., 2022; Zhu et al., 2022; Ghorbel et al., 2023; Liu et al., 2023), modules (e.g., skip connection and attention) (Cheng et al., 2020; He et al., 2022), and activation functions (e.g., generalized divisive normalization) (Ballé et al., 2016a) were introduced. Meanwhile, for better entropy models, hyperprior entropy models (e.g., mean-scale hyperprior) (Ballé et al., 2018; Minnen et al., 2018; Kim et al., 2022) and context models (e.g., spatial autoregressive and channel-wise autoregressive) (Minnen et al., 2018; Lee et al., 2019; Minnen & Singh, 2020) were proposed.

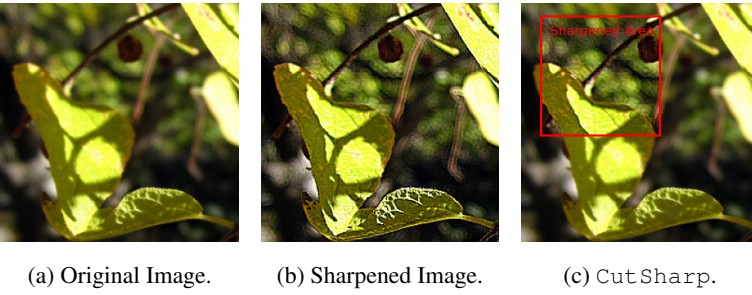

(a) Original Image.     (b) Sharpened Image.     (c) `CutSharp`.

Figure 1: Example of `CutSharp`.

However, the importance of *data*, which is a fundamental component in the realm of deep learning, has been regrettably overlooked in the study of LIC. Additionally, there is no commonly used dataset for training LIC models. During the recent meeting of JPEG-AI community (JPEG-AI, 2023), the issue of common training and testing data was highlighted as a critical concern. Although Wang et al. (2022) indirectly addressed data-dependent LIC by introducing a data-dependent neural-syntax representation and modifying the structure of analysis/synthesis transformations, they did not directly engage in data-centric research, such as examining how training data impacts performance of LIC models.

In this paper, we scrutinize the impact of training data on joint rate–distortion optimization and propose a straightforward data augmentation method for LIC. To our knowledge, our study is the first *direct* data-centric approach to LIC. Our primary contributions are outlined as follows:

- We motivate the importance of effective data utilization for LIC by showing that models trained on a specific subset of dataset can achieve superior performance to those trained on the entire dataset. Specifically, we bridge the relationship between rate–distortion performance and block-wise RGB standard deviation (B-RGB-SD) of the training dataset. **(Section 4)**
- We propose a simple augmentation method for LIC called `CutSharp`, based on B-RGB-SD analysis. `CutSharp` enhances image sharpness within a certain region during training. We compare our method with existing augmentation techniques and show the superiority of our approach for LIC. (**Section 5**)
- We show that `CutSharp` leads to consistent rate–distortion performance improvement across diverse entropy models (including hyperpriors and context models). Especially, with `CutSharp`, trained models can reconstruct images using a smaller BPP, keeping high-frequency details. **(Section 6)**.

## 2 RELATED WORK

**Learned Image Compression (LIC).** For learned nonlinear transformations, the structure consisting of linear convolutional layers with nonlinear activation functions was first used (Ballé et al., 2017). Unlike other computer vision tasks, generalized divisible normalization (GDN) (Ballé et al., 2016a), which is especially designed for density modeling of images, has been predominantly used as the activation function. More advanced techniques, such as deep neural networks with skip connection and attention modules, have been proposed (Cheng et al., 2020; He et al., 2022). Transformer-based nonlinear transformations have also been adopted (Zhu et al., 2021; Zou et al., 2022). Most recently, an approach based on the fusion of CNN and Transformer exhibited more improved rate–distortion performance (Liu et al., 2023).

As one of the early attempts for learned entropy models, Ballé et al. (2017) assumed that the channels of the latent representation follow different probability models and that all elements in a channel are independent and identically distributed. Based on this assumption, they proposed the factorized entropy model, which does not change according to the input. Other subsequent methods aimed to design adaptive probability models for better rate–distortion performance. To achieve adaptation, information is used, which can be categorized into two groups: side information extracted by

additional neural networks (Ballé et al., 2018), and context information obtained from previously encoded/decoded elements (Minnen et al., 2018). Recent research has focused on how to efficiently extract and exploit information. For instance, Kim et al. (2022) efficiently modeled global side information using a cross-attention mechanism, while other works proposed efficient modeling of context information by improving parallelism (Minnen & Singh, 2020; He et al., 2021).

Some prior works have proposed instance-adaptive compression (Li et al., 2018; van Rozendaal et al., 2021; Wang et al., 2022). Among them, Wang et al. (2022) recently introduced a neural data-dependent transform. The main idea is to generate not only a general latent representation but also a neural-syntax representation through an analysis transformation, and to use this data-dependent neural-syntax representation for an improved synthesis transformation. Although they addressed the data dependency, they primarily focused on modifying the analysis/synthesis transformations. Unlike these prior works, we directly investigate the impact of data on LIC models.

**Data Augmentation.** Data augmentation is an effective method for mitigating overfitting and achieving better performance in diverse research fields, including computer vision (Shorten & Khoshgoftaar, 2019) and NLP (Feng et al., 2021). For vision tasks, simple techniques such as horizontal and vertical flipping, cropping, color jittering (including brightness, contrast, saturation, and hue), and blurring–sharpening can be used to manipulate a single image. Additionally, data augmentation techniques such as MixUp (Zhang et al., 2017), CutMix (Yun et al., 2019), PuzzleMix (Kim et al., 2020), and Manifold MixUp (Verma et al., 2019), which involve mixing two images or latent representations, have been proposed.

These mixing-based augmentation techniques have also inspired new approaches for low-level vision tasks: CutBlur for super-resolution tasks and CutDepth for monocular depth estimation tasks. CutBlur (Yoo et al., 2020) involves cutting a low-/high-resolution image patch and pasting it onto the corresponding high-/low-resolution image region, allowing models to learn both how and where to super-resolve an image. CutDepth (Ishii & Yamashita, 2021) involves cutting a depth image patch and pasting it onto the corresponding RGB image region, which reduces the latent distance between the depth and the RGB images. Inspired by these augmentation methods, we propose a straightforward augmentation method for LIC, named `CutSharp`.

## 3 EXPERIMENTAL SETUP

Our main objective is to investigate the impact of *data* on rate–distortion optimization. To this end, we construct LIC methods using the well-known structures and train them from scratch under the same training setting.

**Models.** We train four LIC methods having different entropy models: Joint hyperprior and context model, Joint (Minnen et al., 2018), channel-wise autoregressive model, ChARM (Minnen & Singh, 2020), unevenly grouped channel-wise autoregressive model, UChARM (He et al., 2022), and global side information model, Informer (Kim et al., 2022). All methods have the same structure of the analysis/synthesis transformations as in the ELIC-sm (He et al., 2022) model. It is noted that unless otherwise mentioned, the Joint method is used.

**Training.** We use 300K images randomly selected from the training images of the Open Images dataset (Krasin et al., 2017). For rate–distortion optimization described in Eq. (1), we utilize mean squared error (MSE) as the distance metric $d$. Following the base code provided by CompressAI (Bégaint et al., 2020), the $\lambda$ values in Eq. (1) consist of $\{0.0018, 0.0035, 0.0067, 0.0130, 0.0250, 0.0483\}$. As $\lambda$ increases, the quality of the reconstructed images improves. For simplicity, we map the $\lambda$ values to reconstruction quality $q$, where the smallest $\lambda$ value corresponds to $q$ of 1 and the largest $\lambda$ value corresponds to $q$ of 6. All models are trained for 100 epochs using the Adam optimizer (Kingma & Ba, 2015). We use a batch size of 16 with randomly cropped patches having a resolution of $256 \times 256$. The learning rate is initially set to $1e^{-4}$ and decreases to $1e^{-5}$ at epoch 90.

**Evaluation.** To measure rate and distortion, we compute the BPP (Bit Per Pixel) and PSNR (Peak Signal-to-Noise Ratio), respectively. We employ BD-rate (Bjøntegaard Delta) (Bjontegaard, 2001) to perform comparisons that take both into account simultaneously. For BD-rate, we first calculate sample-wise BD-rate values, and then average them, following Zhu et al. (2022). The comparison

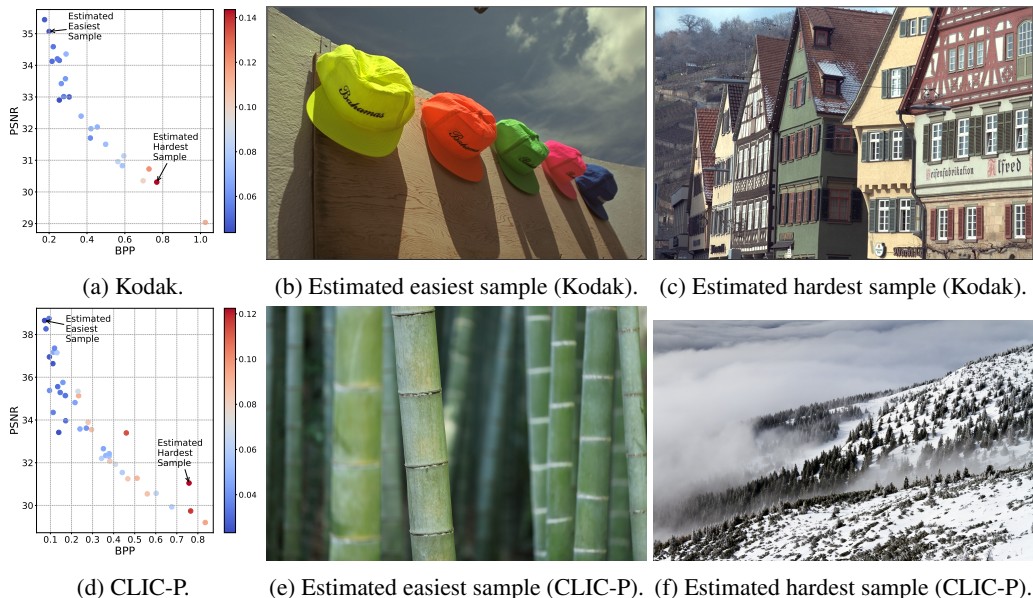

(a) Kodak.  (b) Estimated easiest sample (Kodak).  (c) Estimated hardest sample (Kodak).

(d) CLIC-P.  (e) Estimated easiest sample (CLIC-P).  (f) Estimated hardest sample (CLIC-P).

Figure 2: Rate–distortion performance with block-wise RGB standard deviation of test samples on the (a) Kodak and (d) CLIC-P validation datasets. Each circle represents each test sample. (b) and (e) are the estimated easiest images, and (c) and (f) are the estimated hardest images in each dataset.

baselines are specifically described in each experiment. It should be noted that our comparisons revolve primarily around the presence or absence of certain aspects for LIC. We use the Kodak dataset (Kodak, 1993) and the CLIC professional (CLIC-P) validation dataset (Toderici et al., 2020). The Kodak dataset contains 24 images with resolutions of either $512 \times 768$ or $768 \times 512$. The CLIC-P dataset contains 41 images with an average resolution of approximately $1200 \times 1800$.

## 4  MOTIVATION: AN EFFICIENT DATA-CENTRIC STRATEGY FOR LIC

In this section, we demonstrate that models trained on a subset of dataset, based on B-RGB-SD, can outperform models trained on the entire dataset. This highlights the potential for an efficient data-centric strategy in training LIC models.

**Block-wise RGB Standard Deviation (B-RGB-SD).**  First of all, we analyze the test samples in Kodak and CLIC-P, one by one, using LIC models trained on the entire training dataset. Figures 2(a) and (d) show the rate–distortion performance, where each dot represents one test image, and the BPP and PSNR are averaged over the results from six LIC models according to the reconstruction quality $q$ from 1 to 6.[1] The results reveal a clear linear correlation between BPP and PSNR, with Pearson correlation coefficients of $-0.9394$ and $-0.8988$ for Kodak and CLIC-P, respectively. This means that the number of images with low PSNR at low BPP or high PSNR at high BPP is inherently small when using LIC models. Based on this observation, we categorize images with high PSNR at low BPP as *easier* samples (located in the upper-left area), while images with low PSNR at high BPP as *harder* samples (located in the lower-right area).

Herein, we introduce *block-wise RGB standard deviation* (B-RGB-SD) as a measure for estimating the compression-related difficulty of an image before any training. B-RGB-SD of an image is calculated by dividing it into a set of blocks, calculating the RGB standard deviations for each block, and averaging them.[2] Compared with image-wise RGB-SD, B-RGB-SD is more effective in capturing the compression-related difficulty of an image. Let us consider two $256 \times 256$ images: one that has $128 \times 256$ filled with black and the remaining $128 \times 256$ filled with white, and another that alternates between black and white both horizontally and vertically like a checkerboard. Intuitively, the latter

---

[1]Results without averaging are provided in Appendix C.

[2]We use a block size of $16^2$ because images are typically reduced by $16^2 \times$ via an analysis transformation.

image is thought to be harder to compress than the former image. Notwithstanding the fact that the two images have the same image-wise RGB-SD, the latter image has much higher B-RGB-SD than the former image. In Appendix A, we provide samples with image-wise and block-wise RGB-SD.

With this concept, in Figures 2(a) and (d), stronger red indicates higher B-RGB-SD, while stronger blue indicates lower B-RGB-SD. As expected, blue dots are generally located in the upper-left easier area, whereas red dots are generally located in lower-right harder area. Moreover, Pearson correlation coefficients are calculated to provide further evidence. For Kodak and CLIC-P, the coefficients between B-RGB-SD and BPP are $0.8957$ and $0.7739$, respectively, and the coefficients between B-RGB-SD and PSNR are $-0.7989$ and $-0.6400$, respectively. This indicates that there is a strong correlation between B-RGB-SD and compression-related difficulty. For instance, Figures 2(b) and (e) are the images with the lowest B-RGB-SD, implying that they are expected to be among the easiest images to compress by LIC models. Conversely, Figures 2(c) and (f) are images with the highest B-RGB-SD, implying that they are expected to be among the hardest images to compress. Although previous research has acknowledged the connection between the complexity (or, difficulty) of an image and its local color characteristics (e.g., contrast) (Zou et al., 2022), our approach is distinct in that we explicitly use B-RGB-SD as an metric analyze its relationship with the performance of LIC models.

**Training Dataset Based on the B-RGB-SD.** Beyond the analysis on the test samples, we examine the influence of the B-RGB-SD of the *training* dataset on the rate–distortion performance. Specifically, we order the images in the training dataset based on their B-RGB-SD and divide them into five mutually exclusive groups. Although images are cropped during training, the overall trend of B-RGB-SD is still maintained. The hardest training subset includes the images with the highest B-RGB-SD, which correspond to the top 0-20% of the dataset, while the easiest training subset includes the images with the lowest B-RGB-SD, which correspond to the top 80-100% of the dataset. For fair comparison, we use the same number of train-

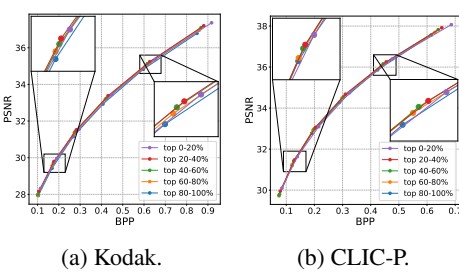

(a) Kodak.    (b) CLIC-P.

Figure 3: Rate–distortion performance according to B-RGB-SD of training dataset.

ing steps for all cases. Namely, when using a subset (20%) of the dataset, the models are trained for 500 epochs, because when using the entire dataset, the models are trained for 100 epochs.

Figure 3 shows the rate–distortion performance according to the estimated compression-related difficulty (i.e., B-RGB-SD) of the training dataset. The results reveal that on average, as the B-RGB-SD increases (i.e., blue → orange → green → red → purple), the LIC models tend to target higher PSNR at higher BPP. This indicates that even though the rate–distortion training objectives are the same, the difficulty of the training dataset can affect the desired PSNR and BPP. The detailed values of BPP and PSNR are provided in Appendix E.

Next, we provide an assessment of the trade-off between rate and distortion by calculating the BD-rate for four samples (shown in Figure 2) compared to models trained on the entire dataset, as depicted in Figure 4. The lower PSNR at lower BPP achieved due to the easier training dataset is beneficial for the easiest test samples (Figures 4(a) and (c)), whereas the higher PSNR at higher BPP achieved due to the harder training dataset is beneficial for the hardest test samples (Figures 4(b) and (d)). This indicates that the trade-off between rate and distortion is influenced by the difficulty of both the training and test samples.

Lastly, we evaluate these models on the entire test samples, ranging from the easiest sample to the hardest sample on average. Table 1 shows the BD-rate compared to the model trained on the entire dataset. It is observed that trained only using a subset of the dataset, LIC models can outperform those trained using the entire training dataset, on both Kodak and CLIC-P. This indicates that possessing a greater number of samples does not necessarily result in better performance, even when models are trained with an unlabeled dataset. Additionally, it seems that the optimal performance can be achieved at a distinct sweet spot for each test dataset, rather than following the same optimal direction. This finding motivates the importance of considering the B-RGB-SD when training LIC models.

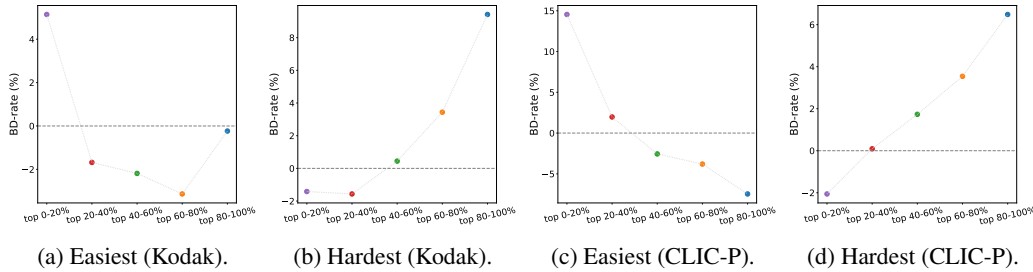

(a) Easiest (Kodak).  (b) Hardest (Kodak).  (c) Easiest (CLIC-P).  (d) Hardest (CLIC-P).

Figure 4: BD-rate compared to the model trained on the entire dataset. Evaluation is performed on four samples: (a) Estimated easiest sample in Kodak (Figure 2(b)), (b) Estimated hardest sample in Kodak (Figure 2(c)), (c) Estimated easiest sample in CLIC-P (Figure 2(e)), and (d) Estimated hardest sample in CLIC-P (Figure 2(f)).

| Test Dataset | Training Dataset (Average of B-RGB-SD) | | | | | |
|---|---|---|---|---|---|---|
| | Entire (0.0656) | top 0-20% (0.1092) | top 20-40% (0.0776) | top 40-60% (0.0618) | top 60-80% (0.0482) | top 80-100% (0.0311) |
| Kodak | 0.00% | 1.88% | **-1.20%** | -0.82% | -0.20% | 2.60% |
| CLIC-P | 0.00% | 4.31% | 0.17% | -0.61% | **-0.82%** | 0.24% |

Table 1: BD-rate compared to the model trained on the entire dataset. Lower is better.

## 5 CutSharp: Enhancing Sharpness within a Certain Region

Although it is observed that utilizing a subset of the training dataset can be advantageous for training LIC models, constructing a suitable subset based on B-RGB-SD is challenging. This is because the training dataset for the LIC task can vary considerably and the appropriate range varies depending on test dataset as shown in Table 1. Consequently, to ensure generalizability, we aim to use the given entire dataset with an augmentation technique. This converts the subset selection problem into the manipulation of the B-RGB-SD of the training dataset. In light of this, we propose a novel augmentation technique, named `CutSharp`.

### 5.1 Color-related Augmentation Techniques

We first examine the impact of the two representative color-related augmentation techniques (i.e., ColorJitter and Blurring–Sharpening, provided by PyTorch (Paszke et al., 2019)) on rate–distortion performance, considering changes in B-RGB-SD. To investigate the impact, we vary the magnitude $m(> 0)$ of these augmentation techniques. ColorJitter changes an image's brightness, contrast, and saturation to a random degree between the minimum and maximum values. If both the minimum and maximum values are equal to $1.0$, the transformed image is identical to the original one. For a more comprehensive analysis, we divide ColorJitter into two subgroups: negative ColorJitter, in which the minimum value is defined as $1.0 - m$ and the maximum value is fixed at $1.0$, and positive ColorJitter, in which the minimum value is fixed at $1.0$ and the maximum value is defined as $1.0 + m$. Blurring–Sharpening changes an image's sharpness based on the given degree of value. Blurring–Sharpening is divided into two subgroups: Blurring and Sharpening with the degree of $m$. In this experiment, the probability of augmentation is set to one.

Table 2 presents the B-RGB-SD values according to the magnitudes of ColorJitter and Blurring–Sharpening, and the BD-rate compared to the models trained without any augmentation. As expected, negative ColorJitter and Blurring decrease the B-RGB-SD of the training dataset, whereas positive ColorJitter and Sharpening increase the B-RGB-SD of the training dataset. The magnitude $m$ is strongly correlated with the degree of increase or decrease in the B-RGB-SD of the training dataset. The change in B-RGB-SD resulting from ColorJitter is greater than that resulting from Blurring–Sharpening.

Regarding the rate–distortion performance, these data augmentation techniques typically do not result in improvements for LIC models. Instead, for each augmentation technique, the BD-rate tends

| Augmentation ($m$) | B-RGB-SD | BD-rate ($\downarrow$) | | Augmentation ($m$) | B-RGB-SD | BD-rate ($\downarrow$) | |
| --- | --- | --- | --- | --- | --- | --- | --- |
| | | Kodak | CLIC-P | | | Kodak | CLIC-P |
| neg. ColorJitter (0.8) | 0.0235 | 5.92% | 5.64% | Blurring (1.00) | 0.0585 | 36.48% | 27.72% |
| neg. ColorJitter (0.6) | 0.0319 | 3.85% | 3.34% | Blurring (0.75) | 0.0601 | 12.77% | 8.67% |
| neg. ColorJitter (0.4) | 0.0419 | 1.88% | 1.07% | Blurring (0.50) | 0.0618 | 3.99% | 2.44% |
| neg. ColorJitter (0.2) | 0.0531 | 0.73% | -0.63% | Blurring (0.25) | 0.0637 | 1.06% | 0.45% |
| – | 0.0656 | 0.00% | 0.00% | – | 0.0656 | 0.00% | 0.00% |
| pos. ColorJitter (0.2) | 0.0761 | 1.84% | 3.40% | Sharpening (0.25) | 0.0681 | -0.39% | 0.16% |
| pos. ColorJitter (0.4) | 0.0835 | 4.76% | 7.17% | Sharpening (0.50) | 0.0702 | -0.46% | 0.42% |
| pos. ColorJitter (0.6) | 0.0888 | 9.43% | 12.24% | Sharpening (0.75) | 0.0723 | 0.39% | 1.76% |
| pos. ColorJitter (0.8) | 0.0928 | 12.08% | 15.42% | Sharpening (1.00) | 0.0747 | 0.88% | 2.34% |

Table 2: B-RGB-SD according to the magitude $m$ of ColorJitter and Blurring–Sharpening and BD-rate compared to the model trained without any augmentation on Kodak and CLIC-P.

to decrease initially and then increase, as the B-RGB-SD increases from the minimum to the maximum. Intriguing observations are that the change in BD-rate is asymmetric with respect to the magnitude change, and the change in BD-rate varies depending on the augmentation technique. This result implies that there are other characteristics affected by augmentation other than B-RGB-SD, as evidenced by the comparison between negative ColorJitter and Blurring. Among augmentation techniques, Sharpening exhibits the lowest rate–distortion performance degradation for both Kodak and CLIC-P datasets.

## 5.2 Algorithm

Based on our previous experiment, we adopt Sharpening and introduce `CutSharp`, illustrated in Figure 1. Our method cuts a sharpened image patch and pastes it onto the corresponding original image region. To implement `CutSharp`, we select a random region following CutMix (Yun et al., 2019), limit the size of this region to a maximum size of $s$, and sharpen the restricted region with a magnitude of $m$. The intention behind `CutSharp` is related to mimicking the property of the original image during training, wherein a smaller cropped image is used. In other words, during training, the original image is cropped to a smaller size, which is more likely to have a low B-RGB-SD because typically only a small portion of the original image has high B-RGB-SD. Therefore, to compensate this imbalance, we apply Sharpening to an arbitrary part of the cropped image during training, to increase the B-RGB-SD.

## 5.3 Ablation Study

We conduct an ablation study on the hyperparameters of `CutSharp`: magnitude $m$ and maximum size $s$. Table 3 presents the BD-rates obtained when applying `CutSharp`, compared to the models without augmentation. It is observed that `CutSharp` can improve BD-rate for both the Kodak and CLIC-P datasets (bold texts in Table 3), although the improvement is slight. The improvement on both datasets is not achieved when using existing augmentation techniques, as shown in Table 2. Furthermore, it is observed that strong magnitude with a small maximum size or weak magnitude with a large maximum size are effective for `CutSharp`.

| $m$ \ $s$ | $8 \times 8$ | $16 \times 16$ | $32 \times 32$ | $64 \times 64$ |
| --- | --- | --- | --- | --- |
| 0.5 | 0.01% | 0.11% | -0.38% | **-0.24%** |
| 1.0 | -0.14% | 0.33% | -0.39% | -0.13% |
| 1.5 | **-0.34%** | -0.18% | -0.35% | -0.23% |

(a) Kodak.

| $m$ \ $s$ | $8 \times 8$ | $16 \times 16$ | $32 \times 32$ | $64 \times 64$ |
| --- | --- | --- | --- | --- |
| 0.5 | 0.26% | 0.31% | -0.08% | **-0.40%** |
| 1.0 | 0.25% | 1.13% | 0.18% | 0.40% |
| 1.5 | **-0.21%** | -0.14% | -0.16% | -0.14% |

(b) CLIC-P.

Table 3: Ablation study on hyperparameters, magnitude $m$ and maximum size $s$.

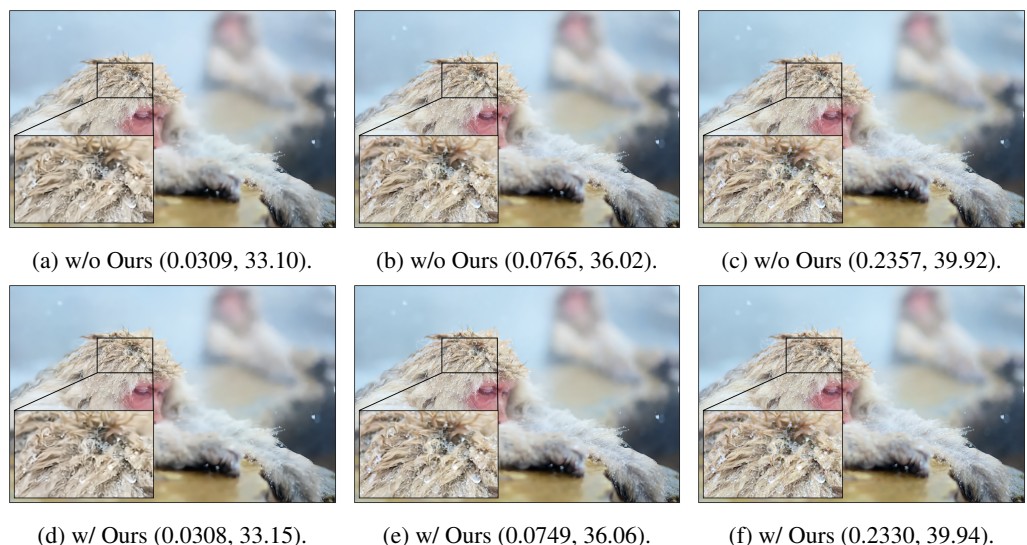

(a) w/o Ours (0.0309, 33.10).    (b) w/o Ours (0.0765, 36.02).    (c) w/o Ours (0.2357, 39.92).

(d) w/ Ours (0.0308, 33.15).    (e) w/ Ours (0.0749, 36.06).    (f) w/ Ours (0.2330, 39.94).

Figure 5: Qualitative comparison on the test sample, on which the largest performance gain appears. Top and bottom rows indicate the reconstructed images by without and with `CutSharp`, respectively. Three reconstruction qualities $q$ are used: 1 for (a) and (d), 3 for (b) and (e), and 6 for (c) and (f). BPP and PSNR are reported in parentheses.

## 6 EXPERIMENTS

For this section, we set the magnitude $m$ to 0.5 and the maximum size $s$ to $64 \times 64$ for `CutSharp`.

### 6.1 QUALITATIVE RESULTS

We compare the visual quality of decoded images depending on `CutSharp` on the sample image from CLIC-P, on which the largest BD-rate gain appears. Figure 5 displays three comparisons according to the reconstruction quality $q$ with (BPP, PSNR) values: 1 for (a) and (d), 3 for (b) and (e), and 6 for (c) and (f). Our results demonstrate that the model with `CutSharp` can reconstruct this sample similar to the model without `CutSharp` while using a smaller BPP, maintaining details (zoomed box).

### 6.2 RATE–DISTORTION PERFORMANCE ACROSS VARIOUS MODELS

Table 4 presents the BD-rate achieved with `CutSharp` across various LIC methods. The comparison baselines of the models using `CutSharp` are the counterparts that do not use `CutSharp`. For instance, the baseline of Informer with `CutSharp` is Informer without `CutSharp`.

Although the performance improvement is minor, we highlight consistent enhancements in the rate–distortion performance across diverse LIC methods. This finding implies that data-centric LIC and model-centric LIC can be developed together to achieve improved performance for LIC. By leveraging both approaches, we can potentially create more robust and efficient LIC models.

| Model | CutSharp | BD-rate (↓) Kodak | BD-rate (↓) CLIC-P |
|---|---|---|---|
| Joint (2018) | ✗ | 0.00% | 0.00% |
| | ✓ | -0.24% | -0.40% |
| ChARM (2020) | ✗ | 0.00% | 0.00% |
| | ✓ | -0.46% | -0.18% |
| Informer (2022) | ✗ | 0.00% | 0.00% |
| | ✓ | -0.66% | -0.48% |
| UChARM (2022) | ✗ | 0.00% | 0.00% |
| | ✓ | -0.42% | -0.33% |

Table 4: BD-rate according to `CutSharp` across diverse LIC methods.

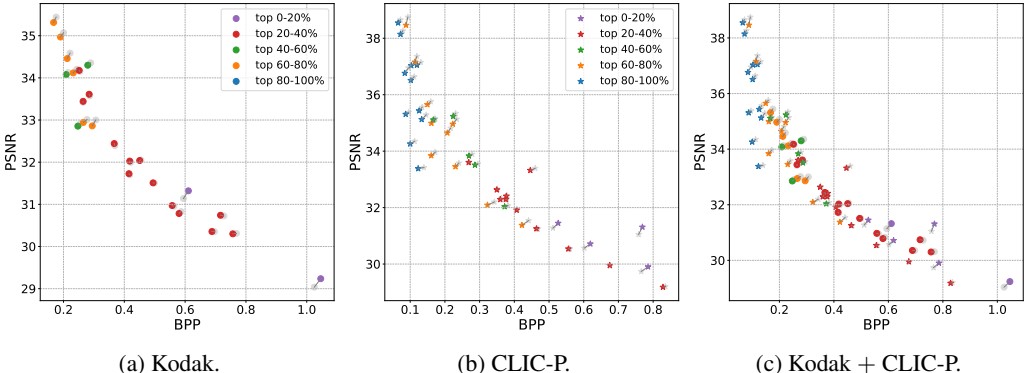

Figure 6: Changes in rate–distortion performance of test samples on (a) Kodak (circle marker), (b) CLIC-P (star marker), and (c) Together. Gray dots represent the performance of the model trained using the entire training dataset, while colored dots represent the performance of the best model among the models trained using a subset of training dataset.

## 7 CONCLUSION

We motivate data-centric approaches for LIC by showing that using the entire training dataset does not guarantee the better performance. For data analysis, we introduce B-RGB-SD as a tool to estimate the compress-related difficulty of an image. We conduct an in-depth sample-wise analysis using this measure, and propose a simple data augmentation method for LIC. Our proposed augmentation technique demonstrates a consistent improvement in rate-distortion performance, albeit slight. We hope that our research will encourage further exploration of data-centric approaches for LIC, and open up new opportunities in this area.

### 7.1 DISCUSSION

We discuss the effectiveness of model ensemble for LIC by examining the match between the B-RGB-SD of the training dataset and that of the test dataset. To assess the potential of model ensemble, we extend the observations presented in Figure 4 for all test samples, and select the best model for each test sample based on BD-rate among the five models that are designed in Section 4. Figure 6 depicts the changes in the averaged BPP and PSNR for each test sample, comparing the performance of the model trained using the entire dataset to that of the best model trained using a subset of the training dataset. The gray dots in Figures 6(a) and (b) are the same as those in Figure 2(a) and (d), respectively. The colored dots represent the best performance among the five models, and their color denotes the selected model. The lines connect dots that correspond to the same sample. Additionally, we present the combined results for Kodak and CLIC-P in Figure 6(c), using circle and star markers, respectively.

In Figures 6(a) and (b), there is a gradual color shift (i.e., the best model among five models) from blue to orange, orange to green, green to red, and red to purple, from the upper-left to the lower-right, although the ratio of each color is not precisely 20%. Furthermore, from Figure 6(c), when the two datasets are analyzed simultaneously with B-RGB-SD, this trend becomes more apparent: models trained using easier training dataset tend to perform better on easier test samples (i.e., blue or orange dots are located in the upper-left region), while models trained using harder training dataset tend to perform better on harder test samples (i.e., purple or red dots are located in the lower-right region). This implies that B-RGB-SD is an effective measure of approximating the compress-related difficulty of an image, and that matching the B-RGB-SD between the training and test datasets is an important factor in improving performance of LIC models.

Finally, from Figures 6(a) and (b), the BD-rates of ensembled models are $-1.57\%$ and $-2.27\%$ on Kodak and CLIC-P, which is better than the BD-rates of the best non-ensembled model ($-1.20\%$ and $-0.82\%$), listed in Table 1. Although we select the best model for discussion purposes, this result highlights the potential of an ensemble model or a mixture of experts (Rincy & Gupta, 2020; Ganaie et al., 2022), based on the characteristics of the training and test samples.

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

## A IMAGE-WISE AND BLOCK-WISE RGB-SD EXAMPLES

Table 5 displays image samples from Kodak, along with their image-wise RGB-SD (I-RGB-SD) and block-wise RGB-SD values. The BPP and PSNR values are from Figure 2(a). As mentioned in the main paper, it is worth noting that B-RGB-SD has a much stronger correlation with BPP and PSNR than I-RGB-SD. For instance, the 20th sample has large I-RGB-SD (0.3463), but small B-RGB-SD (0.0529). This sample is deemed to be easily compressible. Red and blue texts indicate the estimated harest and easiest samples based on B-RGB-SD.

| Image | | | | | |
|---|---|---|---|---|---|
| I-RGB-SD | 0.1560 | 0.0921 | 0.1715 | 0.1614 | 0.1898 |
| B-RGB-SD | 0.1000 | 0.0448 | 0.0438 | 0.0550 | 0.1210 |
| BPP | 0.6965 | 0.2539 | 0.2002 | 0.2775 | 0.7276 |
| PSNR | 30.35 | 32.90 | 35.07 | 33.01 | 30.72 |
| I-RGB-SD | 0.2280 | 0.1542 | 0.2437 | 0.1483 | 0.1506 |
| B-RGB-SD | 0.0733 | 0.0709 | 0.1436 | 0.0537 | 0.0547 |
| BPP | 0.4987 | 0.2895 | 0.7687 | 0.2218 | 0.2434 |
| PSNR | 31.51 | 34.36 | 30.31 | 34.59 | 34.20 |
| I-RGB-SD | 0.1674 | 0.1787 | 0.2096 | 0.1913 | 0.3205 |
| B-RGB-SD | 0.0700 | 0.0515 | 0.1132 | 0.0843 | 0.0612 |
| BPP | 0.4210 | 0.2152 | 1.0249 | 0.5633 | 0.2644 |
| PSNR | 32.00 | 34.13 | 29.04 | 30.96 | 33.42 |
| I-RGB-SD | 0.1668 | 0.1895 | 0.1516 | 0.1868 | 0.3463 |
| B-RGB-SD | 0.0484 | 0.0625 | 0.0761 | 0.0693 | 0.0529 |
| BPP | 0.3062 | 0.2862 | 0.5868 | 0.3689 | 0.2540 |
| PSNR | 33.00 | 33.58 | 30.83 | 32.39 | 34.15 |
| I-RGB-SD | 0.1755 | 0.1726 | 0.2116 | 0.2105 | |
| B-RGB-SD | 0.0702 | 0.0594 | 0.0475 | 0.0828 | |
| BPP | 0.4544 | 0.4178 | 0.1759 | 0.5944 | |
| PSNR | 32.05 | 31.70 | 35.44 | 31.14 | |

Table 5: Image characteristics of the Kodak dataset.

Moreover, we provide B-RGB-SD histogram with RGB distribution for Kodak dataset. Figure 7 describes the process of B-RGB-SD calculation, where the last scalar value is used for representing B-RGB-SD in our paper. In Table 6, the upper line indicates pixel-level RGB distribution and the lower line B-RGB-SD histogram, corresponding to each image in Table 5. For RGB distributions, minimum and maximum values of x-axis are 0 and 255, and those of y-axis are 0 and 50,000. For B-RGB-SD histogram, minimum and maximum values of x-axis are 0 and 0.40, and those of y-axis are 0 and 500. The B-RGB-SD histogram of most samples is concentrated at low values.

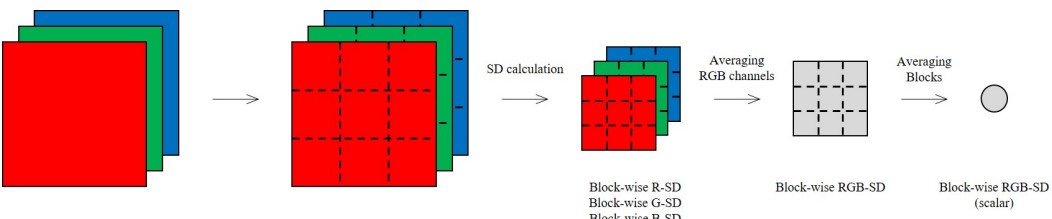

Figure 7: Overview of B-RGB-SD calculation.

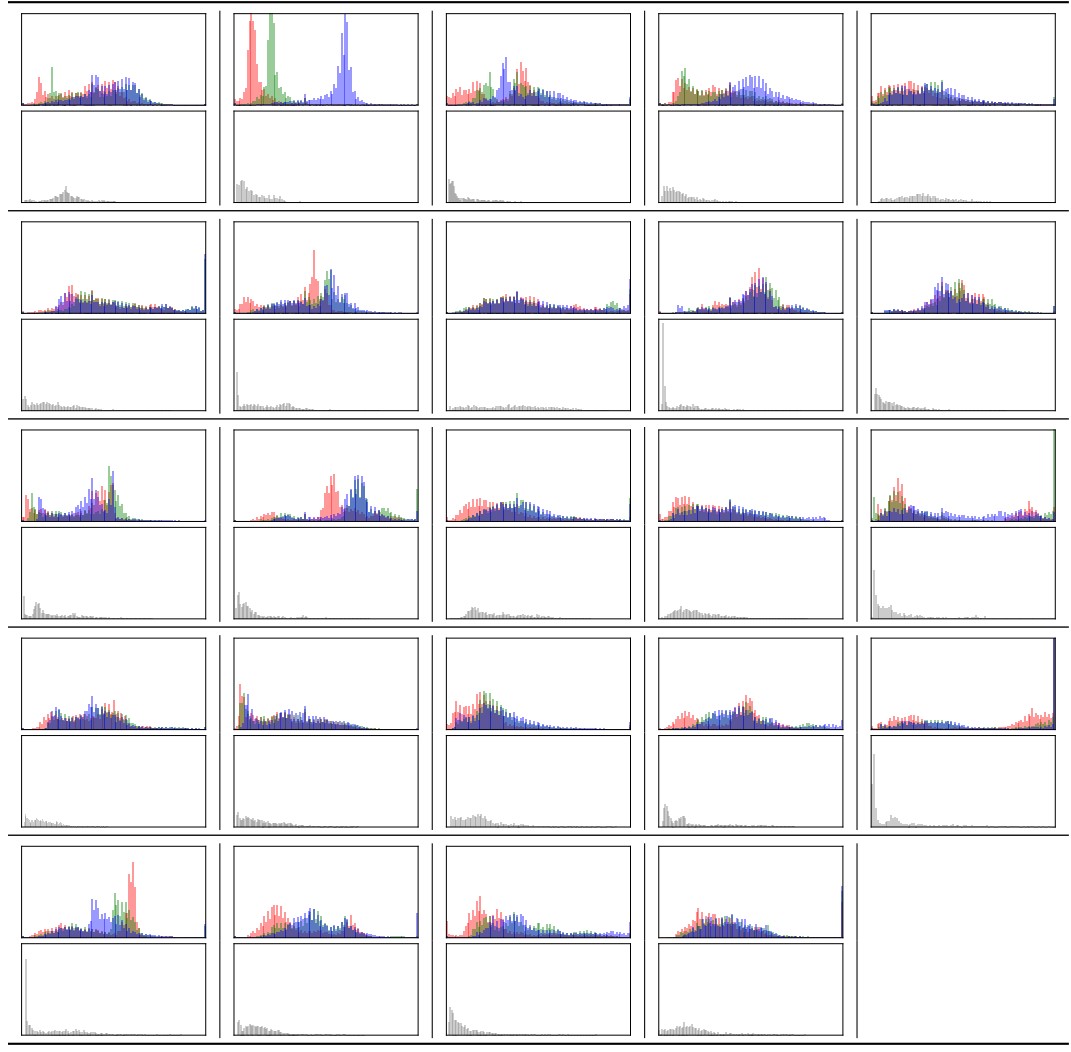

Table 6: RGB distributions and B-RGB-SD histogram of the Kodak dataset.

## B    Correlation between B-RGB-SD According to the Block Size

In the main paper, we used a block size of $16^2$ to compute the B-RGB-SD of an image. However, the B-RGB-SD values may vary depending on the block size. In particular, the ranking of these values is crucial because we divided the Open Images dataset into five groups based on B-RGB-SD, to ensure the validity of our results. Therefore, we calculate two types of correlation, Pearson correlation and Spearman rank correlation, as shown in Table 7. This result demonstrates that our results hold true if the block size does not exceed a certain threshold (e.g., $128^2$).

|  | $4^2$ | $8^2$ | $16^2$ | $32^2$ | $64^2$ | $128^2$ |
|---|---|---|---|---|---|---|
| $4^2$ | 1.00 | 0.98 | **0.93** | 0.84 | 0.72 | 0.58 |
| $8^2$ | 0.98 | 1.00 | **0.98** | 0.92 | 0.81 | 0.68 |
| $16^2$ | **0.93** | **0.98** | **1.00** | **0.98** | **0.90** | 0.79 |
| $32^2$ | 0.84 | 0.92 | **0.98** | 1.00 | 0.97 | 0.88 |
| $64^2$ | 0.72 | 0.81 | **0.90** | 0.97 | 1.00 | 0.96 |
| $128^2$ | 0.58 | 0.68 | 0.79 | 0.88 | 0.96 | 1.00 |

(a) Pearson correlation.

|  | $4^2$ | $8^2$ | $16^2$ | $32^2$ | $64^2$ | $128^2$ |
|---|---|---|---|---|---|---|
| $4^2$ | 1.00 | 0.99 | **0.94** | 0.86 | 0.75 | 0.61 |
| $8^2$ | 0.99 | 1.00 | **0.98** | 0.93 | 0.83 | 0.70 |
| $16^2$ | **0.94** | **0.98** | **1.00** | **0.98** | **0.91** | 0.80 |
| $32^2$ | 0.86 | 0.93 | **0.98** | 1.00 | 0.97 | 0.89 |
| $64^2$ | 0.75 | 0.83 | **0.91** | 0.97 | 1.00 | 0.96 |
| $128^2$ | 0.61 | 0.70 | 0.80 | 0.89 | 0.96 | 1.00 |

(b) Spearman rank correlation.

Table 7: Correlation between B-RGB-SD of images on the Open Images dataset, according to the block size.

## C    Rate–distortion Analysis of Test Samples

Figure 8 shows the BPP and PSNR for all test samples on Kodak and CLIC-P, across reconstruction quality $q$ from 1 to 6, without averaging them. Each gray line represents the same sample and can be viewed as the rate–distortion curve for that sample. In this figure, when comparing the samples with the same BPP, the samples with higher PSNR can be considered easier than the samples with lower PSNR. Following each gray line, if the samples have the same reconstruction quality $q$, then the easier samples are located in the upper-left region. For example, on Kodak, when the BPP is 0.25, one sample has a PSNR of 24.0 with $q$ of 1, while another sample has a PSNR of 38.0 with $q$ of 5. Then, the latter one can be considered simpler than the former one. When the latter's reconstruction quality reaches 1, following its rate–distortion curve, it is located in the upper-left region compared to the former. This finding is consistent with our definition of easier and harder samples on average, as explained in Figures 2(a) and (d).

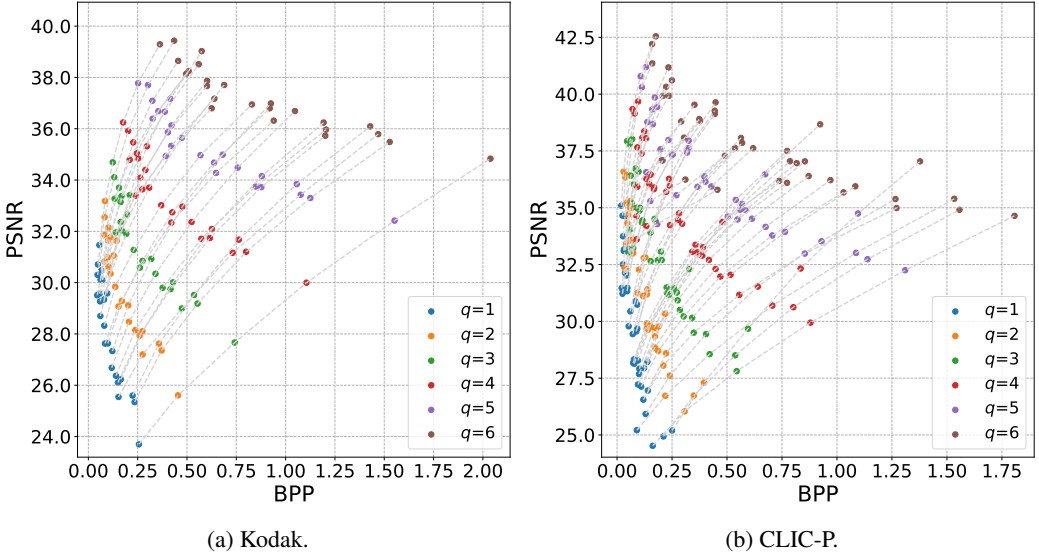

(a) Kodak.    (b) CLIC-P.

Figure 8: Rate–distortion performance on test samples. Each dotted gray line means the same sample according to the reconstructed quality $q$.

# D   B-RGB-SD WHEN TRAINING AND TESTING

To ensure the feasibility of our analysis, we use a B-RGB-SD for each image. Nonetheless, when training models, an image is cropped, resulting in multiple B-RGB-SDs. This does not allow us to divide the entire dataset into the five disjoint groups, designed in Section 4. Instead, we present the B-RGB-SD statistics for the five models designed during training. Table 8 provides the basic statistics of B-RGB-SD when training and testing. For B-RGB-SD statistics during training, only one epoch is processed. The mean value trends are generally similar, although there is overlap in the interval.

|      | Entire | top 0-20% | top 20-40% | top 40-60% | top 60-80% | top 80-100% |
|------|--------|-----------|------------|------------|------------|-------------|
| Mean | 0.0728 | 0.1169    | 0.0858     | 0.0695     | 0.0552     | 0.0366      |
| Std. | 0.0381 | 0.0357    | 0.0274     | 0.0246     | 0.0224     | 0.0199      |
| Max  | 0.3778 | 0.3515    | 0.2844     | 0.2538     | 0.2042     | 0.1918      |
| Min  | 0.0000 | 0.0000    | 0.0000     | 0.0000     | 0.0000     | 0.0000      |

(a) Training.

|      | Entire | top 0-20% | top 20-40% | top 40-60% | top 60-80% | top 80-100% |
|------|--------|-----------|------------|------------|------------|-------------|
| Mean | 0.0656 | 0.1092    | 0.0776     | 0.0618     | 0.0482     | 0.0311      |
| Std. | 0.0285 | 0.0202    | 0.0053     | 0.0040     | 0.0039     | 0.0074      |
| Max  | 0.3394 | 0.3394    | 0.0877     | 0.0690     | 0.0550     | 0.0412      |
| Min  | 0.0008 | 0.0877    | 0.0690     | 0.0550     | 0.0412     | 0.0008      |

(b) Testing.

Table 8: Statistics of B-RGB-SD of the Open Images dataset when training and testing.

# E   DETAILS OF BPP AND PSNR

| Training Dataset | BPP/PSNR($q=1$) | BPP/PSNR($q=2$) | BPP/PSNR($q=3$) | BPP/PSNR($q=4$) | BPP/PSNR($q=5$) | BPP/PSNR($q=6$) |
|------------------|-----------------|-----------------|-----------------|-----------------|-----------------|-----------------|
| Entire     | 0.1072/28.15 | 0.1803/29.85 | 0.2875/31.48 | 0.4339/33.33 | 0.6325/35.22 | 0.8863/37.18 |
| top 0-20%  | 0.1136/28.32 | 0.1907/29.96 | 0.2958/31.56 | 0.4488/33.45 | 0.6532/35.37 | 0.9174/37.36 |
| top 20-40% | 0.1057/28.16 | 0.1770/29.80 | 0.2825/31.52 | 0.4312/33.38 | 0.6271/35.24 | 0.8806/37.20 |
| top 40-60% | 0.1007/27.99 | 0.1741/29.69 | 0.2764/31.42 | 0.4221/33.24 | 0.6153/35.13 | 0.8677/37.10 |
| top 60-80% | 0.1011/27.94 | 0.1682/29.56 | 0.2721/31.35 | 0.4170/33.16 | 0.6099/35.02 | 0.8593/36.96 |
| top 80-100%| 0.1035/27.94 | 0.1681/29.42 | 0.2674/31.15 | 0.4071/32.93 | 0.5968/34.83 | 0.8490/36.78 |

(a) Kodak.

| Training Dataset | BPP/PSNR($q=1$) | BPP/PSNR($q=2$) | BPP/PSNR($q=3$) | BPP/PSNR($q=4$) | BPP/PSNR($q=5$) | BPP/PSNR($q=6$) |
|------------------|-----------------|-----------------|-----------------|-----------------|-----------------|-----------------|
| Entire     | 0.0786/29.93 | 0.1306/31.54 | 0.2056/33.04 | 0.3112/34.65 | 0.4575/36.25 | 0.6552/37.90 |
| top 0-20%  | 0.0847/30.10 | 0.1392/31.63 | 0.2150/33.09 | 0.3249/34.75 | 0.4765/36.39 | 0.6863/38.07 |
| top 20-40% | 0.0785/29.95 | 0.1288/31.47 | 0.2044/33.07 | 0.3098/34.68 | 0.4550/36.25 | 0.6526/37.92 |
| top 40-60% | 0.0741/29.75 | 0.1263/31.41 | 0.1986/32.98 | 0.3025/34.55 | 0.4437/36.15 | 0.6395/37.82 |
| top 60-80% | 0.0744/29.72 | 0.1210/31.25 | 0.1945/32.92 | 0.2974/34.48 | 0.4371/36.06 | 0.6283/37.69 |
| top 80-100%| 0.0764/29.80 | 0.1205/31.20 | 0.1907/32.77 | 0.2886/34.28 | 0.4246/35.86 | 0.6154/37.56 |

(b) CLIC-P.

Table 9: BPP and PSNR according to the training dataset.

