# OpenReview forum: "CutSharp: A Simple Data Augmentation Method for Learned Image Compression"
_ICLR.cc/2024/Conference — Submitted to ICLR 2024_

### Official Review · Reviewer_VyXU · 2023-10-14

**Soundness:** 3 good
**Presentation:** 4 excellent
**Contribution:** 2 fair
**Rating:** 3
**Confidence:** 4

**Summary:**

## Summary
This paper study the data aspect of learned image compression. An interesting analysis using different subset of dataset shows that the performance of learned image compression is effected by the difficulty of images. Motivated by this, the authors propose a novel data augumentation approach for image compression, which brings an improvement of BD-BR of 0.66\%.

**Strengths:**

## Strength
* This paper discusses an unexplored area of learned image compression.
* The method proposed by this paper is well motivated and the analysis using different subsets of dataset is convincing.
* This paper is well-written and clearly presented.

**Weaknesses:**

## Weakness
* First of all, I am not sure if data augumentation is really necessary for image compression at the first place. Image compression is an unsupervised task, and the data is very very easy to acquire. Any image dataset can be directly used. And there are aboundant large datasets, such as Openimage (9,000k) and LAION-5b (5,000,000k), that have not been used by image compression at all. Instead of performing data augumentation on a small dataset (the authors use 300k subset of Openimage), directly use a larger dataset seems to be the most straightforward way. Afterall, natural image dataset are already very very large. For example, the authors train the model with 90 epochs on 300k images, which means that any image dataset with 27,000k is enough to provide a different image for each iter. And obvious candidates for this task would include LAION-400M (400,000k), LAION-5B high res (170,000k) and LAION-5B (5,000,000k).
* Second, the empirical results is too weak to justify that the data augumentation. The best improvement for BD-BR is -0.66\%, which is too marginal. For example, a -5.0\% improvement in BD-BR is approximately 0.2 dB improvement in BD-PSNR. And a BD-BR of -0.66\%, without reporting standard deviation, is too weak to support a paper.

**Questions:**

## Questions
* It would be much more convincing if the authors are dealing with special images that are hard to acquire. For example, medical image, remote sensing image. Currently the authors only test their methods on natural image, which is too cheap to acquire and eliminates the need for data augumentation.
* The current empirical results is too weak, and the authors should provide an improvement of 2\% to 5\% improvement of BD-BR to support a paper, this might be achieved by chaning the domain of image.
* A perhaps more intuitative measure of difficulty is the distance between single image's R-D and average R-D curve. This can be obtained by any pre-trained image codec. It would be interesting to see whether this achieves the same effect as B-RGB-SD.

---

> ### Author Response · Authors · 2023-11-16
> **Response to Reviewer VyXU**
>
> We would like to express our gratitude for your thoughtful comments.
>
> In order to address any concerns you have raised, we have itemized the weaknesses and questions and provided our responses below.
>
> ### W1. Necessity of data augmentation for unsupervised tasks
> - We believe that data augmentation is necessary to efficiently use the same dataset, even for unsupervised tasks.
> - In this vein, CutSharp consistently improves performance compared to no data augmentation (albeit slight), when using the same training dataset.
> - [Please refer to Table 1] Furthermore, we showed that just collecting data is not the answer, because the performance trained using top40-60% dataset (size: 60K) is better than the performance trained using entire dataset (size: 300K). It implies that large dataset size does not guarantee higher performance for LIC.
>
> ### W2. Slight improvement of CutSharp
> - [Please refer to Table 4] We admit that CutSharp did not bring much performance improvement. Nevertheless, please note that CutSharp is an augmentation technique with consistent improvement across various LIC models.
> - We will update the standard deviation of the results for 3 runs.
>
> ### Q1&Q2. Specific-domain where acquiring data is hard
> - Thank you for the insightful comment! However, we think that this comment focuses on ``domain'' for LIC, which is slightly different from the direction our paper is aiming for.
> - This is because we addressed B-RGB-SD as an intrinsic characteristic of an image (i.e., sample-wisely), but domain is an intrinsic characteristic of a cluster of images.
> - In addition, we should design four settings for addressing this comment:
>   - Setting1. training dataset - natural (i.e., acquiring data is easy), test dataset - natural
>   - Setting2. training dataset - natural (i.e., acquiring data is easy), test dataset - speical
>   - Setting3. training dataset - speical (i.e., acquiring data is hard), test dataset - natural
>   - Setting4. training dataset - speical (i.e., acquiring data is hard), test dataset - speical
> - Although the comparison between above settings would be greatly impactful, it is likely to be a new work. Rather, we would like to state that through our work, data augmentation is generally necessary for LIC.
>
> ### Q3. Distance between single image's RD and averaged RD curve, instead of B-RGB-SD.
> - [Please refer to Section 4 (Figure 2)] We would like to correlate compression difficulty (e.g., bpp/psnr after training LIC) with the intrinsic characteristics of the image before training (i.e., codec-independently). This is because by doing this, we could design augmentation techniques of images.
> - Furthermore, the calculation time for (codec-independent) B-RGB-SD is much smaller than the calculation time for (codec-dependent) RD-curve.
> - [Please refer to Appendix C (Figure 8)] Nevertheless, we provided single image's RD curves of both Kodak and CLIC-P in Appendix C. In this figure, if the RD curves corresponding to the samples are located below, then it can be inferred that those samples are easier. It means that the distance you suggested has a similar role to B-RGB-SD.
>
> We hope our response could address your questions.

---

> > ### Comment · Reviewer_VyXU · 2023-11-17
> >
> > Thanks for the reply.
> >
> > Currently, the major issue is that I am still not convinced that data augumentation for training codec of natural image is an issue, given the quite minimal improvement by the proposed approach.
> >
> > Currently the authors argue that using top 50 percent (60K) of dataset can achieve same effect of using the whole. While the problem is, there are already 5,000,000K dataset available. May be one can not achieve better result using x2 data, but one can choose to use x1,000 data. The natural image data is extremely cheap. Alone with the minimal better result, I am not convinced that data augumentation for training codec for natural image of natural image is an issue.
> >
> > This paper is acceptable to me if the authors can show that their approach work for dataset that is expensive to curate, e.g., medical image / remote sensing image / screen content. Despite the authors give the plan, no actual evaluation is conducted. Then I could not determine whether the proposed approach actually works.
> >
> > Calculating codec-based B-RGB-SD can be very cheap, like insN mentions, even JPEG is usable. JPEG has hardware implementation and it can be even cheaper than the proposed approach.
> >
> > Overall, I keep my rating.

---

### Official Review · Reviewer_insN · 2023-10-25

**Soundness:** 2 fair
**Presentation:** 3 good
**Contribution:** 1 poor
**Rating:** 3
**Confidence:** 5

**Summary:**

The paper discusses three things
1) How to measure "difficulty" of an image
2) Use the above measure to train on a subset of the data according to its difficulty
3) Use an augmentation called CutMix

**Strengths:**

It is interesting that CutSharp improves BD-Rate marginally.

**Weaknesses:**

Part 1: Difficulty of an image. Authors propose using RGB standard deviation of RGB blocks to get a measure for the difficulty of an image, and show in Fig 2. how this correlates to bpp/PSNR. They report a person correlation. The weakness lies in no baselines given: if we have a codec (eg even just JPEG) we can trivially calculate bpp/PSNR and with two or more images we could fit a line. How does that compare to the proposed method?

Part 2: Table 1 does not show an actionable insight. The proposed B-RBG-SD is apparently not enough to select a subset, because the optimal subset is only known after trainign a model on each. I.e. The authors cannot come up with a prediction of what a useful subset is, so the method is meaningless. In Fig.4, we also don't see a novel insight: training on hard examples improves hard inference, but makes easy images worse, and vice-versa, this is just classical ML: you are good where you train.

Part 3: CutSharp: The augmentation provides very miniuscule gains only (Table 4) and in Fig 5 we see no obvious visual diff (not surprising given the tiny PSNR diffs, e.g., 0.05dB between a) and d).

Overall, this paper does not contain insights or results that warrant an ICLR publication.

**Questions:**

Generally, people use augmentation because labelling is hard. In image compression, one can just get more data, no labels required. It would have been insightful to double the training set and see how that affects BD-rate gains. I'd be surprised if CutSharp gets bigger gains than you get from doubling the dataset.

---

> ### Author Response · Authors · 2023-11-16
> **Response to Reviewer insN**
>
> We would like to express our gratitude for your thoughtful comments.
>
> In order to address any concerns you have raised, we have itemized the weaknesses and questions and provided our responses below.
>
> ### W1. No baselines of B-RGB-SD
> - [Please refer to Section 4 (Figure 2)] We aimed to establish a correlation between compression difficulty (e.g., bpp/psnr after training LIC) and the intrinsic characteristics of the image before training (i.e., codec-independently). This is important because it would enable us to design image augmentation techniques.
> - [Please refer to Appendix A] With this intention, we introduced Image-wise RGB-SD (I-RGB-SD) and Block-wise RGB-SD (B-RGB-SD) for the Kodak dataset. As we mentioned in Section 4, the Pearson correlations between B-RGB-SD and bpp/psnr are 0.8957/-0.7989. However, the Pearson correlations between I-RGB-SD and bpp/psnr are 0.0692/0.0023.
> - For comparison, a measure that has a stronger correlation with compression performance is better. Therefore, B-RGB-SD is a better measure than I-RGB-SD.
> - Furthermore, we calculate the Pearson correlations between block-wise entropy (calculated by skimage.measure.shannon_entropy) and bpp/psnr. The values are 0.7521/-0.7682, which are weaker than the correlations between B-RGB-SD and bpp/psnr. (will be updated.)
>
> ### W2. Utilization of B-RGB-SD
> - [Please refer to Table 1 and Figure 4] From the results, we would like to highlight that B-RGB-SD is a pretty accurate measure to estimate compression-related difficulty, rather than strong actionable message.
> - As you mentioned, "training on *easy* examples improves *easy* inference, and training on *hard* examples improves *hard* inference" is undoubted.
> - However, without the use of "B-RGB-SD," we could not explicitly demonstrate this hypothesis for LIC.
> - [Please refer to Section 5] Therefore, we proposed a new augmentation method by considering B-RGB-SD of training dataset, inspired by the fact that a small portion of an image has relatively higher B-RGB-SD than the rest of it, in general.
>
> ### W3. Slight improvement of CutSharp
> - [Please refer to Figure 5 and Table 4] We admit that there is not much difference in quality comparison because CutSharp did not bring much performance improvement. Nevertheless, please note that CutSharp is an augmentation technique that consistently improves the performance of various LIC models.
>
> ### Q1. Necessity of data augmentation for unsupervised tasksd
> - We agree that collecting data is not difficult for unsupervised tasks.
> - [Please refer to Table 1] However, we demonstrated that just increasing dataset size is not the answer, because the performance trained using top40-60% dataset (size: 60K) is better than the performance trained using entire dataset (size: 300K). Rather, we suggest B-RGB-SD should be considered, even for collecting data for LIC.
> - Furthermore, data augmentation like CutSharp can improve performance even when using the same dataset.
> - In addition, we will update the results of CutSharp using random 50% of the Open Images dataset, following your comments.
>
>
> We hope our response could address your questions.

---

> > ### Comment · Reviewer_insN · 2023-11-17
> >
> > Thanks for your reply.
> >
> > Regarding W1: Thanks for calculating the shannon entropy over blocks.
> > Regarding W2: What I mean with "actionable message" is that you had to train a model on each subset. To decide what is best, you had to use the validation set (Kodak). That just means you overfitted to the validation set. I think in the rebuttal you write that without B-RGB-SD you could not have split the dataset. So if your contribution is "we find a method that split the dataset such that when we look at the validation performance we can find a split that is better", you would need to add a baseline method to show that what you do is useful. For example, what if you take a random 20% of the data with 5 different seeds? Are you expecting a big gain?
> > Regarding W3: It did not "consistently improve results", as we see in Table 3, eg m=0.5, s = 16x16 made results worse. Like above, you had to tune the parameter of the augmentation on the dataset you evaluate on to get better results. The proper way would be to use a holdout set of the training data to tune the parameters.
> >
> > Q1: You did not demonstrate that increasing the dataset helps, you just showed that selecting a subset by looking at validation helps.
> >
> > I will stick with my rating.

---

### Official Review · Reviewer_p52F · 2023-10-27

**Soundness:** 2 fair
**Presentation:** 2 fair
**Contribution:** 2 fair
**Rating:** 3
**Confidence:** 5

**Summary:**

The paper studies the problem of data augmentation for learned image compression and develops a data-centric method for this problem based on the block-wise RGB standard deviation, called CutSharp. The method aims to balance B-RGB-SD by enhancing image sharpness in the training phase. Based on CutSharp, the learned image compression achieves better rate-distortion performance. The paper presents superior results across various learned image compression methods on the Kodak and CLIC datasets.

**Strengths:**

1. CutSharp is a simple and direct data-centric approach that utilizes data effectively in learned image compression.

2. CutSharp leads to consistent rate–distortion performance improvement across diverse learned image compression.

**Weaknesses:**

1. My major concern is the limited technical novelty and contribution of the paper. CutSharp is a simple idea but just a variant of the sharpening method -- sharpen the random region of an image. It compensates for the imbalance of B-RGB-SD of the cropped image.

2. The experiments in the paper are not convincing and the overall performance of CutSharp is not strong enough. First of all, the improvement of CutSharp on Kodak (-0.39%) and CLIC (-0.40%) is worse than Sharpening=0.50 on Kodak (-0.46%) and ColorJitter=0.2 on CLIC (-0.63%).  The visualization of learned motion patterns in Fig. 5 seems not appealing enough.

3. The analysis of B-RGB-SD is not convincing enough. The experiments reveal that B-RGB-SD is more effective in capturing the compression-related difficulty of an image. However, it is not clear how B-RGB-SD influences the compression performance.

**Questions:**

1. Sec.5.2 said "In other words, during training, the original image is cropped to a smaller size, which is more likely to have a low B-RGB-SD because typically only a small portion of the original image has high B-RGB-SD." -> How did you come to this conclusion? More experiments are suggested to support this claim.

2. If the claim of Q1 is true, the problem of imbalance on B-RGB-SD may come from the crop process during training. The crop process leads to low B-RGB-SD samples. I am wondering what if you drop the low B-RGB-SD samples?

---

> ### Author Response · Authors · 2023-11-16
> **Response to Reviewer p52F**
>
> We would like to express our gratitude for your thoughtful comments.
>
> In order to address any concerns you have raised, we have itemized the weaknesses and questions and provided our responses below.
>
> ### W1. Technical novelty and contribution
> - [Please refer to our general response] We believe that our work has valuable contributions as the first data-centric approach for LIC.
> - With regard to CutSharp, although we use the basic augmentation technique of "Sharpening" and adopt "Cut" idea, we analyzed existing augmentation techniques for LIC tasks and adopted "Cut" idea based on our B-RGB-SD analysis and intuition.
>
> ### W2. Slight improvement of CutSharp
> - [Please refer to Table 2 and Table 3] We would like to highlight that CutSharp can improve for both datasets, although the improvement is slight. The improvement on both datasets was not achieved when using only one fixed augmentation.
> - [Please refer to Figure 5 and Table 4] We admit that there is not much difference in quality comparison because CutSharp did not bring much performance improvement. Nevertheless, please note that CutSharp is an augmentation technique with consistent improvement across LIC models.
>
> ### W3. Effects of B-RGB-SD to compression performance
> - [Please refer to Figure 3] We showed that on average, as the B-RGB-SD of the training dataset increases (i.e., blue → orange → green → red → purple), the LIC models tend to target higher PSNR at higher BPP, even if objective functions are the same.
> - [Please refer to Figure 4] Next, we provided a sample-wise assessment of the trade-off between rate and distortion by calculating the BD-rate.
>
> ### Q1. B-RGB-SD during training
> - [Please refer to Section 5.2] Firstly, we strongly apologize for providing incorrect information. We would like to modify sentences from "In other words, during training, the original image is cropped to a smaller size, which is more likely to have a low B-RGB-SD because typically only a small portion of the original image has high B-RGB-SD. Therefore, to compensate this imbalance, we apply Sharpening to an arbitrary part of the cropped image during training, to increase the B-RGB-SD." to "Typically, only a small portion of the original image has relatively higher B-RGB-SD than the rest of it, as described in Table 6 in Appendix A. Therefore, to imitate this imbalance, we apply Sharpening to an arbitrary part of the cropped image during training, to increase the B-RGB-SD."
> - [Please refer to Appendix A (Table 6)] We provided B-RGB-SD histogram (before averaging) of Kodak dataset. In general, the histograms of images are negatively-skewed distributions, and this property is the intuition behind CutSharp.
>
> ### Q2. Dropping low B-RGB-SD
> - In Section 5.2, low B-RGB-SD was expressed negatively, but I would like to correct it in Q1 and apologize once again.
> - [Please refer to Table 1] Regardless of Q1, Table 1 provides a rough answer to this question. Using the training dataset with top 40-60% B-RGB-SD brings performance improvement, which implies that training samples with high or low B-RGB-SD can be harmful.
> - We will update the results using different training dataset setups (e.g., top0-80%, top10-90%, top20-80%, etc).
>
>
> We hope our response could address your questions.

---

### Official Review · Reviewer_316K · 2023-11-01

**Soundness:** 2 fair
**Presentation:** 3 good
**Contribution:** 1 poor
**Rating:** 3
**Confidence:** 4

**Summary:**

This paper proposes a simple data augmentation method, called Cutsharp, for learned image compresion problem. Specifically, the authors first introduce block-wise RGB standard deviation as a measure for estimating the compression difficulty of images. Then they demonstrate that compression models trained on a certain subset of images, selected from the entire training set according to the block-wise RGB standard deviation, can achieve superior rate-distortion performance to the models trained on the entire dataset. Inspired by the above observation, the authors propose a data augmentation strategy to boost the peformance of the neural image compression model, called Cutsharp, which enhances image sharpness within an randomly selected region in the training images.

**Strengths:**

- This paper seeks to investigate a new question: how the quality of the training data affects the results of rate-distortion optimization, which has been neglected in previous studies.
- The experiments are comprehensive and some of the findings are very interesting, e.g,. there is a strong correlation between the block-wise RGB standard deviation and the compression difficulty.
- The idea of dividing training images into different groups according to the B-RGB-SD metric is great.

**Weaknesses:**

- The finding  "_models trained on a subset of dataset, based on B-RGB-SD can outperform models trained on the entire dataset_" is meaningless. It is just some kind of __overfitting__.
The test dataset, either Kodak or CLIC, only contain a few dozen images.
When dividing the entire dataset into different groups according to the B-RGB-SD metric, one of the divided subsets happens to be closer to the test set in some of the statistical properties  (e.g, B-RGB-SD). So It is no doubt that the model trained on this subset will lead to a better result on the test set. I think that Figure 4 can only prove that the subset of top 20-40% B-RGB-SD is closer to Kodak than other subsets and the subset of top 60-80% B-RGB-SD is the one closest to CLIC dataset.

- Another major weakness of this paper is lack of motivation. Only the data augmentation method (Cutsharp) is given, but there is no explanation as to why it is designed this way. Why sharpening and why cut?
 -- Why sharpening? There are so many operations for low-level image processing, why choose sharpening?  What is the specificity of the sharpening operation? Why not try the combination of different operations?
 -- Why cut? What will happen if the operation (such as sharpening) performed on the entire image?

**Questions:**

Please see weaknesses.

---

> ### Author Response · Authors · 2023-11-16
> **Response to Reviewer 316K**
>
> We would like to express our gratitude for your thoughtful comments.
>
> In order to address any concerns you have raised, we have itemized the weaknesses and questions and provided our responses below.
>
> ### W1. Overfitting
> - [Please refer to Figure 4] As you mentioned, a smaller distribution shift between the training dataset and test sample undoubtedly leads to performance improvement: training on *easy* examples improves *easy* inference, and training on *hard* examples improves *hard* inference.
> - However, without the use of "B-RGB-SD," we could not explicitly demonstrate this hypothesis for LIC.
> - In other words, B-RGB-SD is a measure that captures compression-related difficulty (i.e., easy or hard), which we have highlighted through our work.
> - [Please refer to Section 5] Therefore, we proposed a new augmentation method by considering B-RGB-SD of training dataset, rather than matching distribution between training and test datasets, because we realized the infeasibility and challenges of matching distributions.
>
> ### W2. Lack of CutSharp's motivation
> - [Please refer to Section 5.1] We described the details and ineffectiveness of four color-related augmentation techniques with B-RGB-SD: positive/negative ColorJitter, Blurring, and Sharpening. Among these techniques, Sharpening exhibits the lowest rate–distortion performance degradation for both Kodak and CLIC-P datasets.
> - [Please refer to Table 2, Table 3, and Table 4] Nevertheless, Sharpening could not improve the rate–distortion performance on both datasets. Therefore, we borrowed the "Cut" idea, inspired by the fact that a small portion of an image has relatively higher B-RGB-SD than the rest of it, in general (as described Table 6). Finally, this leads to consistent improvement.
>
>
> We hope our response could address your questions.

---

> > ### Comment · Reviewer_316K · 2023-11-22
> >
> > Thanks for your reply. I stick with my rating.

---

### Author Response · Authors · 2023-11-16
**General Response**

Dear reviewers,

We greatly appreciate your valuable comments and suggestions in regards to our work.
Firstly, we would like to summarize the main contributions and strengths raised by reviewers. Then, we have addressed weaknesses and questions for each reviewer.

We would like to emphasize our contributions to the learned image compression community.
1. Our work is the first to introduce a direct data-centric approach for LIC.
2. We introduced the Block-wise RGB standard deviation (B-RGB-SD) metric, which has a strong correlation with compression performance (e.g., bpp/psnr) and approximately captures the compression-related difficulty of images before any training.
3. Furthermore, we demonstrated how different training datasets according to B-RGB-SD affect the rate-distortion performances through a sample-wise analysis.
4. Lastly, we proposed a simple data augmentation method based on B-RGB-SD, coined CutSharp. This leads to slight but consistent performance improvement for LIC tasks.

The reviewers have highlighted the following strengths of our work:
- A new and unexplored question for LIC (@316K, @VyXU), and well-motivated and well-written (@VyXU)
- A simple and direct data-centric approach, leading to consistent RD performance improvement (@p52F)
- Interesting findings and comprehensive experiments (@316K)

If you have any further questions or suggestions, please feel free to share them on OpenReview. We are committed to addressing all raised concerns promptly and in accordance with the reviewing policy.

---

### Meta-Review · Area_Chair_uLJA · 2023-12-06

**Metareview:**

This work proposes a new data augmentation approach for improving the compression tradeoff. Authors have a relatively new perspective here wrt to the compression challenge by addressing it via a data-driven approach.

**Justification For Why Not Higher Score:**

The paper lacks motivation or theoretical understanding of why such modification could improve the results.
Moreover, the results are not exhaustive enough for the reviewers to believe that such approach would generalize consistently to other, more real, scenarios.

**Justification For Why Not Lower Score:**

N/A

---

### Decision · Program_Chairs · 2024-01-16

Reject